# A Study of Magnetic Properties in a 2D Ferromagnetic Nanolattice through Computer Simulation

**DOI:** 10.3390/nano12203705

**Published:** 2022-10-21

**Authors:** Sergey V. Belim, Igor V. Bychkov

**Affiliations:** 1Department of Physics, Omsk State Technical University, 644050 Omsk, Russia; 2Department of Electronics and Radiohysics, Chelyabinsk State University, 454001 Chelyabinsk, Russia

**Keywords:** 2D nanolattice, antidote lattice, thin film, phase transition, computer simulation

## Abstract

This article investigated the magnetic properties of a 2D nanolattice through computer modeling. A square antidote nanolattice in thin films was considered. For our computer simulation, we used the Heisenberg model. Ferromagnetic phase transition was studied for lattices with pores of various sizes. We determined the Curie temperature based on the finite-dimensional scaling theory. Using Wolf’s algorithm, we simulated the behavior of the system. The dependence of the phase transition temperature on the density of spins was found to be power. Using Metropolis’ algorithm, we calculated a hysteresis loop for an antidote lattice film. The hysteresis loop narrowed as the pore sizes increased. The dependence of coercive force on the size of the nanolattice obeyed the logarithmic law.

## 1. Introduction

Nanolattices are structured objects with nanometer-sized pores. The pores change the mechanical [1,2], optical [3,4], and magnetic [5,6] properties of the substance. Two-dimensional nanolattices are thin films with holes [7,8]. The holes are placed in the nodes of a two-dimensional periodic matrix. They form an antidote lattice. Nanometer pores change the properties of a thin film: optical [9,10], electrical [11,12] and magnetic [13]. The size and period of the holes control the magnetization and magnetic susceptibility of the film. This effect is used in the development of spintronic devices: magnetic transistors [14,15,16] and magnetic recording media [17]. Antidote lattices increase the surface of the film. Surface effects have greater effects on film properties. There are several technologies for obtaining nanometer pores in thin films: electron beam lithography [18,19], focal ion milling [20], and the self-assembly of nanoscale spheres [9,21,22]. These technologies control the period and pore sizes of antidote lattices.

The behavior of magnetization in a ferromagnetic film with pores is of particular interest. Antidote lattices prevent the movement of magnetic domain walls. This process changes the coercive force [23,24,25,26]. The pores of the lattice also affect the magnetic resistance, magnetic susceptibility, and magnetic anisotropy of the film [27,28,29,30,31,32,33]. The dynamic magnetic properties of the film also vary. The process of magnetic relaxation slows down [34,35,36,37,38,39,40]. Relaxation time can be increased up to 9 times [37] compared with a continuous film. Magnetic domains are fixed at the edges of the holes. Magnetization reversal takes longer. There are different kinds of interactions between spins in ferromagnetic materials. A comparison of micromagnetic modeling results with X-ray microscopy data showed the dominant contribution of exchange interactions to the films with antidote lattices properties [38].

Thin-film-based nanolattices are actively studied experimentally. Thin films of NiFe with antidote lattices change their magnetic properties depending on the size and mutual position of the pores [39]. The antidote lattice has an influence on the process of the magnetization of systems. The remagnetization of Fe, Co, and permalloy films with antidote lattices [40] showed a dependence of hysteresis loop width on linear hole dimensions. The coercive force increases linearly depending on the size of the holes in these systems. The dependence of coercive force on pore sizes and film material was investigated for Fe, Ni, and NiFe [41]. Coercive force linearly depends on the pore size for the lattices of square antidotes. There is also a dependence on the film material. The dependence of magnetic properties on the antidote lattice parameters was investigated for Co films [13,42]. The coercive force depends on the shape and size of the pores and their relative location. Micromagnetic modeling for Co films with antidote lattices was performed for three pore sizes (20, 40, and 60 nm) [43]. The modeling results [43] showed the dependence of the hysteresis loop shape and width on the pore size. Changing hole sizes regulates the magnetic properties of antidote-based nanolattices.

In this study, we performed a computer simulation of ferromagnetic phase transition and film remagnetization with an antidote lattice based on the Heisenberg model.

## 2. Model and Methods

We considered a thin film with a lattice of square antidotes. The pores are located in the nodes of a square grid. The film thickness is D. The pores have a square shape with side b. The lattice period is d. A general view of the film with the antidote lattice is shown in Figure 1.

The system is parallel to the OXY plane. The equations of film surfaces are z=0 and z=D−1.

We used the Heisenberg model to investigate the magnetization behavior of the system. Spin ${\vec{S}}_{i}=\left(S_{ix},S_{iy},S_{iz}\right)$ is matched to each of the film atoms. The spin vector modulus is 1/2 ($\left|{\vec{S}}_{i}\right|=1/2$). A film with an antidote lattice consists of simple ferromagnetic material. An exchange interaction occurs between the spins. The exchange interaction rapidly decreases with distance. Only pairs of nearest neighbors are considered when calculating the Hamiltonian system. The exchange integral for spin interaction is J. The Hamiltonian system includes the sum of the paired exchange interactions for the spins, the term defining the anisotropy of the system, and the term for the interaction of the system with the external magnetic field ${\vec{h}}_{0}$.
(1)$$H=-J\sum {\vec{S}}_{i}{\vec{S}}_{j}+K\sum S_{iz}S_{jz}+\mu_{B}{\vec{h}}_{0}\sum {\vec{S}}_{i}.$$

The first term is responsible for the exchange interaction of the spins. Only near neighbors are present in the summable pairs of spins. The second term includes the anisotropy parameter K. The system is anisotropic along the OZ axis. Anisotropy is present in any ferromagnetic system. The third term is responsible for the interaction with the external magnetic field. The external magnetic field strength is ${\vec{h}}_{0}=\left(h_{0x},h_{0y},h_{0z}\right)$. $\mu_{B}$ is a Bohr magneton. The third term calculates the sum of all the spins.

The system temperature and magnetic field strength are calculated in relative units in computer simulations. If the true temperature of the system is t, then the relative temperature T is measured in terms of the exchange integral.
(2)$$T=\frac{k_{B}t}{J}.$$
where $k_{B}$ is the Boltzmann constant. If the true strength of the magnetic field is ${\vec{h}}_{0}=\left(h_{0x},h_{0y},h_{0z}\right)$, then the relative value is also measured in units of the exchange integral.
(3)$$\vec{h}=\frac{\mu_{B}}{J}\left(h_{0x},\;h_{0y},h_{0z}\right),\;h=\left|\vec{h}\right|.$$

We write the Hamiltonian model in relative quantities for computer modeling.
(4)$$H/J=-\sum {\vec{S}}_{i}{\vec{S}}_{j}+R\sum S_{iz}S_{jz}+\vec{h}\sum {\vec{S}}_{i}.$$

The relative parameter of anisotropy is measured in units of the exchange integral.
(5)R=K/J.

The magnetization of the system is equal to the reduced spin value per node.
(6)$$\vec{m}=\overset{N}{\underset{i=1}{\sum }}{\vec{S}}_{i}/N.$$
where N is the total number of spins in the system.
(7)$$N=\frac{L^{2}}{d^{2}}\left(d^{2}-b^{2}\right)D.$$
where L is the linear dimensions of the system along the OX and OY directions. The periodic boundary conditions along these directions are applied to the system to simulate an infinite system.

Magnetization is an order parameter system in a ferromagnetic phase transition. Determining the phase transition temperature $T_{C}$ depending on the pore size b is an important task. The temperature can be determined based on the finite-dimensional scaling theory [44]. We used the Wolf cluster algorithm [45] to form the thermodynamic configurations of the system. The magnetization value was calculated for each configuration. The thermodynamic average values of values $\langle m^{4}\rangle$ and $\langle m^{2}\rangle$ were calculated from all configurations. We calculated the fourth-order Binder cumulants [46].
(8)$$U_{4}=1-\frac{\langle m^{4}\rangle }{3\langle m^{2}\rangle {}^{2}}.$$

Binder cumulants depend on the temperature and size of the system being modeled. They have the same value for all system sizes at the phase transition point. The $U_{4}$ versus temperature plots for different L system sizes intersect at one point. This point corresponds to the phase transition temperature $T_{C}$.

We also investigated loop hysteresis as a function of antidote lattice parameters. The Metropolis algorithm [47] is used for the simulation of system magnetization in an external magnetic field. The Wolf cluster algorithm destroys information about the previous state of the system and does not show a hysteresis loop.

## 3. Computer Simulation

Through computer simulation, we examined the systems with a lattice antidote period d=16. The film thickness was D=16. We performed a computer experiment on films with different thicknesses. The behavior of the system was qualitatively the same for any film thickness at D≤16. The behavior of the system did not correspond to thin films for thickness D>16. The effects of the antidote lattice were less pronounced for the film thickness of D<16. The influence of pores depends on the spin’s concentration v.
(9)$$v=1-\frac{b^{2}}{d^{2}}.$$

The phase transition temperature and coercive force decreased as the film thickness D decreased. The pore sizes varied from b=0 to b=14 in steps Δb=2. A system with pores of size b = 0 corresponded to a continuous film. The linear dimensions of the system L were selected in proportion to the period of the antidote lattice to meet periodic boundary conditions. The length of the system along one of the axes varied from L=2d to L=6d in increments ΔL=d. The anisotropy parameter was R=0.1. A plot of the Curie temperature $T_{C}$ versus spin concentration v is shown in Figure 2.

The phase transition temperature first rose rapidly with the increasing spin concentration. The Curie temperature growth rate slowed with a pore size equal to half the period (b=d/2). A plot of the phase transition temperature versus spin concentration on a log scale is shown in Figure 3.

It can be seen from the logarithmic plot that $\ln T_{C}$ versus lnv consists of two straight sections. It follows that the dependence of the phase transition temperature on the concentration of spins can be represented by two power functions.
(10)$$T_{C}∼\{\begin{array}{cc} v^{0.47}, & v\leq 0.75,\\ v^{0.24}, & v>0.75. \end{array}$$

The phase transition temperature coincides with the value for the continuous film at v=1. There are no pores in this case.

We investigated the film remagnetization process with an antidote lattice for systems with size L = 64 and periodic boundary conditions. The external magnetic field was directed along the OX axis. We performed a computer experiment with different directions of the magnetic field in the OXY plane. The hysteresis loop changed little when the direction of the magnetic field changed. This result is consistent with the data of other articles for systems with similar symmetry [48,49,50]. The hysteresis loop contracted as the pore sizes decreased. Examples of hysteresis loops for the three pore sizes are shown in Figure 4.

The decrease in hysteresis loop width was associated with a decrease in the total number of spins and the interaction between them. The overall surface of the system increased. The number of spins with fewer than six neighbors increased. It was easier for the magnetic field to turn the spins around. The coercive force also decreased with the increasing pore size. The plot of coercive force $h_{C}$ versus pore size b is shown in Figure 5.

The coercive force decreased following a nonlinear law with the increasing pore sizes. The analysis of the results showed the logarithmic dependence of the coercive force on the wall thickness a=d−b separating the pores. The plot of coercive force $h_{C}$ versus lna is shown in Figure 6.

The coercive force is approximated by the logarithmic function.
(11)$$h_{C}=0.303+\ln\left(d-b\right).$$

A hysteresis loop was present in the magnetization of a 2D nanolattice, as for any ferromagnetic system. The width of the hysteresis loop decreased as the pore sizes increased. The coercive force depends on the logarithmic law of wall thickness between the pores. Our results are well in agreement with the experimental data of previous studies [31,39,40,41,43] and the results of micromagnetic modeling reported in the literature [41].

## 4. Conclusions

Nanolattices based on thin films with pores have magnetic properties different from solid films. We performed computer simulations of a film with an antidote lattice based on the Heisenberg model. The results of the computer simulation were consistent with the experimental data. A magnetic phase transition occurred in thin films with antidote lattice, as in solid films. The change in pore sizes controls the phase transition temperature. The Curie temperature was determined based on the spin concentration and free surface area. The antidote lattice reduced the density of the substance and the concentration of the spins. The pores in the film increased the free surface of the system. The Curie temperature depended on the concentration of the spins in the nanolattice according to the power law. There were two spin concentration intervals with different exponents in the law for the Curie temperature. The phase transition temperature tended to have a value similar to that of a continuous film as the pore sizes decreased.

## Figures and Tables

**Figure 1 nanomaterials-12-03705-f001:**
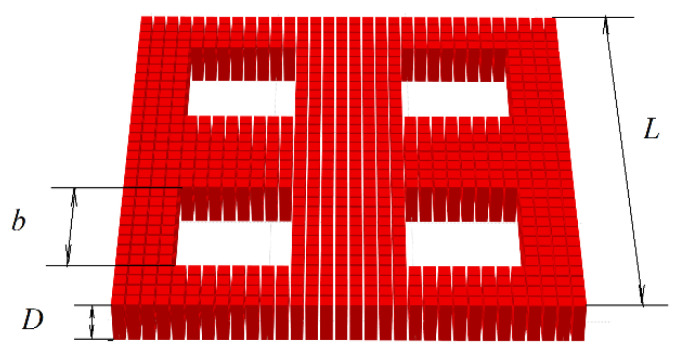
Thin film with antidote lattice.

**Figure 2 nanomaterials-12-03705-f002:**
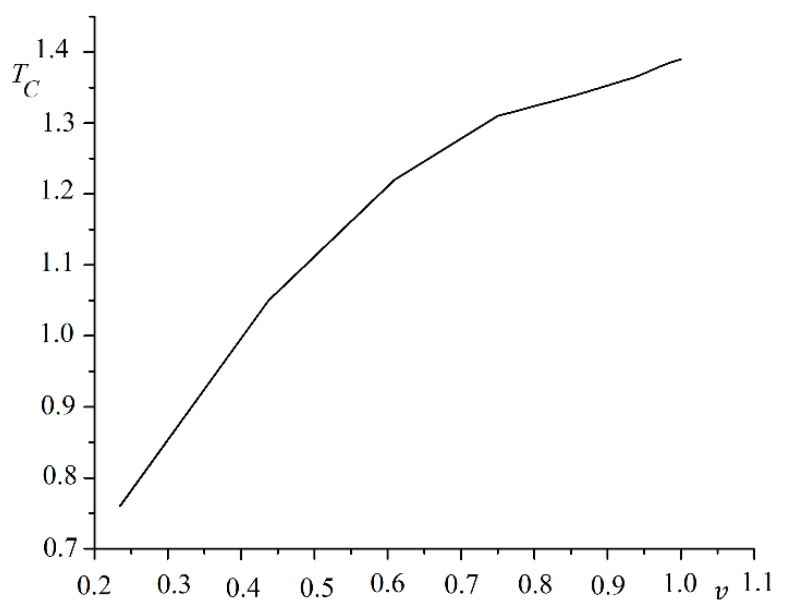
A plot of Curie temperature $T_{C}$ versus spin concentration v.

**Figure 3 nanomaterials-12-03705-f003:**
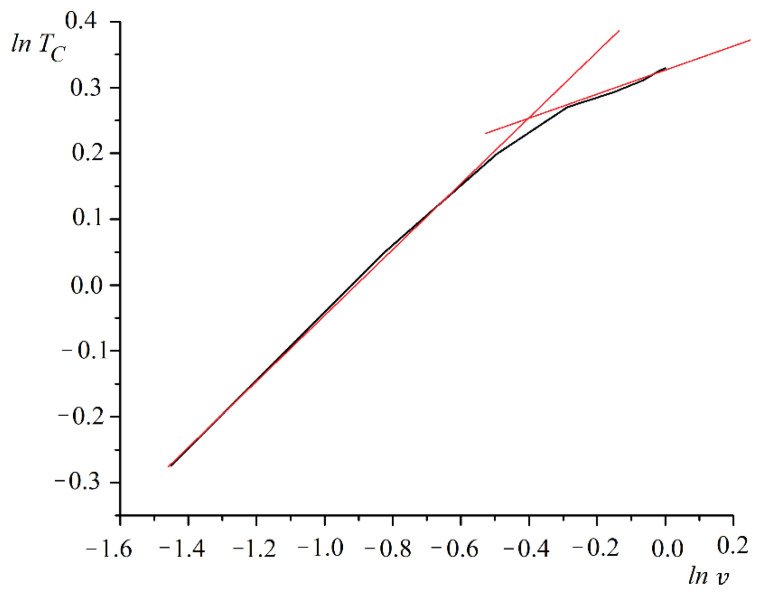
A plot of phase transition temperature $T_{C}$ versus spin concentration v on a log scale.

**Figure 4 nanomaterials-12-03705-f004:**
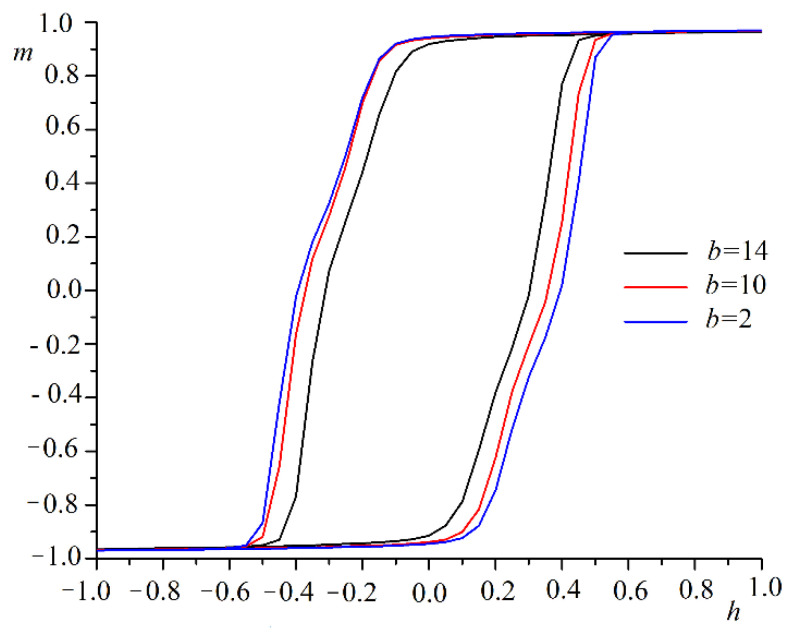
Hysteresis loops for magnetic field along OX axis and different pore sizes b.

**Figure 5 nanomaterials-12-03705-f005:**
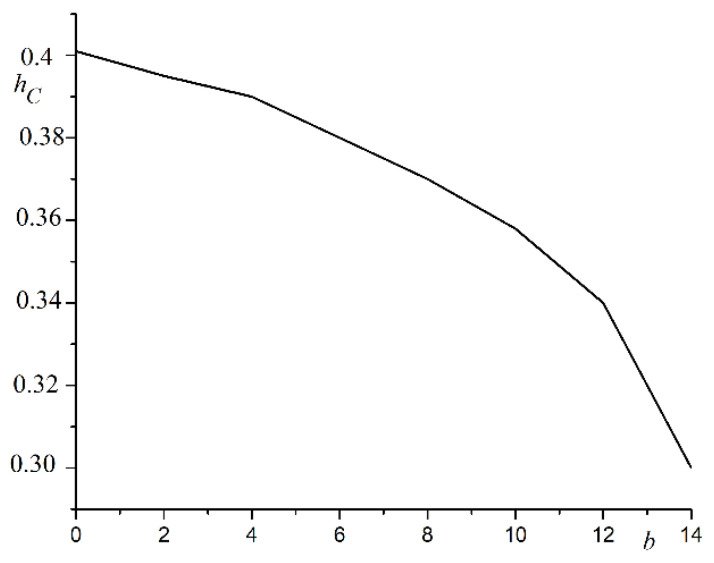
Dependence of coercive force $h_{C}$ on pore size b in magnetic field parallel to OX axis.

**Figure 6 nanomaterials-12-03705-f006:**
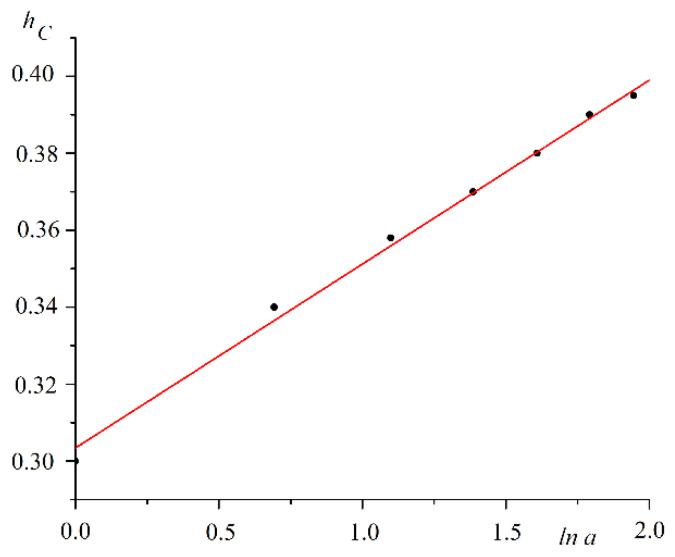
Dependence of coercive force $h_{C}$ on wall thickness between pores a on a semilogarithmic scale.

## Data Availability

Not applicable.

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
