# Peer review of "A Study of Magnetic Properties in a 2D Ferromagnetic Nanolattice through Computer Simulation"

_nanomaterials, 2022, doi:10.3390/nano12203705_

Round 1
Reviewer 1 Report
The manuscript by Belim et al. presents a theoretical study on magnetic properties of 2D ferromagnetic nano-lattice. By computer modeling, the critical temperature of ferromagnetic phase transitions in a thin-film with square antidotes was calculated as a function of the pore size. Moreover, hysteresis loops of the nano-lattice were calculated and the coercive force was extracted. It was claimed that the theoretical results obtained in this work were in agreement with experimental results. Before I can recommend its publication, there are a couple of issues I would like to ask the author to address.
Frist, the theoretical model used in the work includes three energy terms, the Heisenberg exchange energy, anisotropy energy and Zeeman energy. If the magnetostatic energy is also taken into account, how would it affect the results?
Secondly, when the hysteresis loop is calculated, what is the difference between the theoretical model used in the work and the popular micromagnetic simulations? Would the results be the same or different?
Finally, the author may want to improve the qualities of the figures.
Author Response
Dear reviewer!
We thank you for your constructive comments. We provide answers to your questions and comments.
First, the theoretical model used in the work includes three energy terms, the Heisenberg exchange energy, anisotropy energy and Zeeman energy. If the magnetostatic energy is also taken into account, how would it affect the results?
Answer: Consideration of magnetostatic energy requires the introduction of additional dipole-dipole interactions into the Hamiltonian. Dipolar forces are long-range. These forces lead to corrections to the results based on short-range simulations. These adjustments can be small or lead to a qualitative change in the behavior of the system. Accounting for long-range forces is the subject of a separate study. We plan to execute it soon.
Secondly, when the hysteresis loop is calculated, what is the difference between the theoretical model used in the work and the popular micromagnetic simulations? Would the results be the same or different?
Answer: Our results are in good agreement with the results of micromagnetic simulations. However, the Monte Carlo method provides more information on the effect of temperature on the magnetic properties of the system. It takes into account chaotic spin flips that are difficult to account for in micromagnetic modeling. These two modeling methods complement each other.
Finally, the author may want to improve the qualities of the figures.
Answer: We have improved the quality of the figures.

Reviewer 2 Report
Authors have carried out a simulation of 2D ferromagnetic nanolattice and examined the magnetic properties using the Heisenberg model, finite-dimensional scaling theory, Wolf's algorithm, and Metropolis' algorithm. Spin numbers dependent on critical temperature, pore size-dependent Coercive force, and pores' distance-dependent Coercive force was studied. The following points need to be addressed before the acceptance of the manuscript.
1. Authors have not mentioned the framework.
2. Authors have just mentioned the 2D ferromagnetic nanolattice, not said about the specific material on which those models have been implemented.
3. There are many typo errors in the manuscript like lattice is written as sattice and many more manuscripts need to improve throughout for typo errors/ English grammatical mistakes.
4. Remove the last para from the conclusion section and place it at the end of the computer simulation section.
Author Response
Dear reviewer!
We thank you for your constructive comments. We provide answers to your questions and comments.
1. Authors have not mentioned the framework.
Answer: We independently implemented the C++ program and tested it on many simpler models. We believe that this information is not essential to the content of the article.
- Authors have just mentioned the 2D ferromagnetic nanolattice, not said about the specific material on which those models have been implemented.
Answer: Specific materials for antidote nanolattice are indicated in the third paragraph of the introduction. The particular type of material determines the exchange integral J in the model. Our results are universal. Substitution of the exchange integral for a particular substance gives the Curie temperature and coercive force for that material.
- There are many typo errors in the manuscript like lattice is written as sattice and many more manuscripts need to improve throughout for typo errors/ English grammatical mistakes.
Answer: We have corrected errors and typos.
- Remove the last para from the conclusion section and place it at the end of the computer simulation section.
Answer: We removed the paragraph.
